# Changes in Inflammatory Cytokines in Responders and Non-Responders to TNFα Inhibitor and IL-17A Inhibitor: A Study Examining Psoriatic Arthritis Patients

**DOI:** 10.3390/ijms25053002

**Published:** 2024-03-05

**Authors:** Marie Skougaard, Magnus Friis Søndergaard, Sisse Bolm Ditlev, Lars Erik Kristensen

**Affiliations:** 1The Parker Institute, Copenhagen University Hospital Bispebjerg and Frederiksberg, Nordre Fasanvej 57, 2000 Frederiksberg, Denmark; 2Copenhagen Center for Translational Research, Copenhagen University Hospital Bispebjerg and Frederiksberg, Bispebjerg Bakke 23, 2400 Copenhagen, Denmark; 3Department of Clinical Immunology, Aarhus University Hospital, Palle Juul-Jensens Boulevard 99, 8200 Aarhus, Denmark; 4Department of Clinical Medicine, Faculty of Health and Medical Sciences, University of Copenhagen, Blegdamsvej 3B, 2200 Copenhagen, Denmark

**Keywords:** psoriatic arthritis, cytokine, bDMARDs, TNFα, IL-17A

## Abstract

This study aimed to examine the changes in biomarker levels in responders and non-responders to tumor necrosis factor alpha inhibitor (TNFi) and interleukin-17A inhibitor (IL-17Ai) in psoriatic arthritis (PsA) patients over a 4-month period after treatment initiation. A total of 68 PsA patients initiating either TNFi, IL-17Ai, or methotrexate treatment were included. Blood plasma and clinical outcome measures were collected adjacent to treatment initiation and after four months. A commercially available multiplex immunoassay was included to evaluate 54 biomarkers. Mean changes were used to evaluate change over time. A statistically significant decrease in pro-inflammatory cytokines IL-6 (log-transformed mean change −0.97, 95%CI −4.30; 2.37, [*p* = 0.032]) and an increase in anti-inflammatory IL-10 (0.38, 95%CI 1.74; 2.50 [*p* = 0.010]) were seen in TNFi responders. Meanwhile, a statistically significant increase in the target cytokine IL-17A was seen in both IL-17Ai responders (2.49, 95%CI −1.84; 6.85 [*p* = 0.031]) and non-responders (2.48, 95%CI −1.46; 6.41 [*p* = 0.001]). This study demonstrated differing changes in cytokine levels when comparing treatment responders and non-responders, highlighting the need to improve the understanding of the different immune response mechanisms explaining different responses to medical treatment in PsA patients.

## 1. Introduction

Psoriatic arthritis (PsA) is an immune-mediated inflammatory disease characterized by heterogeneous clinical and patient-reported manifestations with a great impact on patients’ quality of life [1]. The introduction of biological disease-modifying anti-rheumatic drugs (bDMARDs) have improved the disease control of arthritis significantly [2]. Further, it has bettered PsA patients’ health-related quality of life [3]. Nonetheless, around 30% of PsA patients have no effect of treatment [4], and the unknown associations between clinical manifestations, immunopathogenesis, and treatment response mechanisms complicate treatment.

PsA has been considered an inflammatory disease of both auto-inflammatory and auto-immune origin [5], constituted by both innate and adaptive immune components [6]. The main theorized mechanisms include the upregulation of the interleukin (IL)-23/IL-17 inflammatory pathway [7], promoted by an abnormal innate response to physical stress [8,9]. The increased levels of pro-inflammatory cytokines are associated with the induction of T cells polarization towards the T helper cell type 17 (Th17) subtype, resulting in IL-17A secretion [10]. However, several additional cell subtypes, cell mediators, and cytokines have been suspected to play an important part in the immune response mechanisms inducing, amplifying, and sustaining the inflammatory mechanisms in PsA, which are the targets of available treatments [6].

Treatments include conventional synthetic disease-modifying anti-rheumatic drugs (csDMARDs), for which pharmacodynamics remain unknown [11], and bDMARDs that target (pro-)inflammatory cytokines known to PsA immunopathogenesis and include treatments such as tumor necrosis factor alpha inhibitor (TNFi), interleukin-17A inhibitor (IL-17Ai), interleukin-23 inhibitor, interleukin-12/23 inhibitor, etc. [12]. Nevertheless, the literature reveals the different effectiveness of individual types of treatment on the diverse manifestations of PsA [13]. The different effects of TNFi and IL-17Ai on PsA disease activity have been associated with different cytokine signatures at baseline, but the significance of the change in biomarkers over time and the association with a clinical outcome remains unclear [14]. The observations indicate the continuous need to improve the understanding of PsA immunopathogenesis and the effect of bDMARDs on inflammation-associated biomarkers in association with different PsA clinical phenotypes.

The primary aim of this study was to assess the effect of medical treatments, TNFi and IL-17Ai, on the levels of 54 cytokines, chemokines, and additional biomarkers in PsA patients stratified by the effect of treatment. The secondary aim was to investigate the associations between biomarkers at baseline and individual clinical- and patient-reported outcomes, as well as the difference in biomarker levels at baseline when comparing patients initiating bDMARDs and methotrexate (MTX), i.e., bDMARD-naïve patients.

## 2. Results

### 2.1. Baseline Characteristics

A total of 68 PsA patients were included in the study: 29 TNFi initiators, 19 IL-17Ai initiators, and 20 MTX initiators. No statistically significant differences were found between the patient treatment groups at baseline when comparing sex, age, or disease duration, or when comparing clinical outcome disease activity scores in PsA (DAPSA), the Psoriasis Area Severity Index (PASI), and VAS patient pain. Statistically significant differences between treatment groups were found when comparing the number of previous bDMARDs and VAS patient fatigue (Table 1). Evaluating the differences between individual groups, a post hoc Dunn’s test with Bonferroni correction revealed a statistically significant difference between IL-17Ai and MTX initiators. When comparing treatment groups, statistically significant differences in biomarker levels at baseline were found in *IL-31*, with higher levels in MTX initiators compared with IL-17Ai initiators; *eotaxin*, with higher levels in MTX initiators compared with TNFi initiators; and *Tie-2*, with higher levels in MTX initiators compared with both TNFi and IL-17Ai initiators. *TNFα* had higher levels in patients initiating IL-17Ai compared with both TNFi and MTX initiators, and *IL-17A* had higher levels in patients initiating IL-17Ai compared with MTX initiators. Additionally, *IL-15* was statistically significantly lower in MTX initiators compared with both TNFi and IL-17Ai initiators (Appendix A).

Twenty-two biomarkers were excluded due to missing data (>10%) caused by biomarkers being below the detection range or by the intra-assay coefficient of variation (CV) being deemed too large. Spearman rank correlations between the remaining 32 biomarkers and 9 PsA disease manifestations revealed no associations, including none of the correlation coefficients reaching ≥0.60, defining the threshold for moderate correlation (Figure 1) [15].

### 2.2. Change in Clinical Outcome Measures in Responders and Non-Responders to Treatment

Follow-up data were obtained from 20 TNFi initiators and 19 IL-17Ai initiators. MTX initiators were only included for the analysis of baseline data due to a low number of patients with follow-up data, limiting additional stratification based on treatment response. Data on PsA patients were only included if they presented with complete clinical data to calculate composite clinical outcome measures. Response to treatment was defined either by a 50% improvement in DAPSA, i.e., DAPSA50, or PASI, i.e., PASI50. Of the PsA patients initiating TNFi, 11 and 9 patients were considered as DAPSA50 responders and non-responders, respectively, while 12 and 6 patients were considered as PASI50 responders and non-responders, respectively. Of the PsA patients initiating IL-17Ai, 7 and 12 were considered as DAPSA50 responders and non-responders, respectively, while 7 and 4 were considered as PASI50 responders and non-responders, respectively (Table 2). Stratifying by the DAPSA50 response demonstrated statistically significant decreases in DAPSA for all treatment responders; however, this was also the case for TNFi DAPSA50 non-responders (mean change −7.16 95%CI −49.90; 35.59 [*p* = 0.039]). Thus, the decrease in DAPSA was higher for TNFi DAPSA50 responders (−28.75 95%CI −81.81; 24.32 [*p* < 0.001]). Stratifying by the PASI50 response demonstrated statistically significant decreases in PASI for TNFi responders (−4.08 95%CI −15.61; 7.45 [*p* = 0.003]) and IL-17Ai responders (−4.97 95%CI −22.95; 13.00 [*p* = 0.016]) (Table 2).

### 2.3. Change in Biomarker Level from Baseline to Follow-Up

Log-transformed biomarker levels were evaluated to assess changes from baseline to follow-up in the TNFi and IL-17Ai initiators, stratifying by response to treatment. Results were visualized by bar charts for TNFi and IL-17Ai DAPSA50 responders versus non-responders (Figure 2 and Figure 3) and for TNFi and IL-17i PASI50 responders and non-responders (Figure 4 and Figure 5). Changes in biomarkers from baseline to the follow-up, presented with numeric values, are available in Appendix A.

#### 2.3.1. Change in Biomarkers in TNFi Responders and Non-Responders (DAPSA50 and PASI50)

Changes in biomarker levels did differentiate over time depending on the DAPSA50 response to TNFi (Figure 2). For DAPSA50 responders, statistically significant log-transformed decreases were seen in CRP (−1.03 95%CI −4.50; 2.44 [*p* = 0.049]), IL-6 (−0.97, 95%CI −4.30; 2.37 [*p* = 0.032]), TARC (−0.629 95%CI −3.446; 2.188 [*p* = 0.042]), and VEGF-D (−0.07, 95%CI −0.98; 0.83 [*p* = 0.042]) and, while statistically significant, increases were demonstrated in IL-12p70 (0.59, 95%CI −2.30; 3.47 [*p* = 0.049]) and IL-10 (0.38, 95%CI −1.74; 2.50 [*p* = 0.009]). DAPSA50 non-responders demonstrated a statistically significant decrease in SAA (−0.33, 95%CI −3.15; 2.48, [*p* = 0.031]) (Appendix A).

Additional results based on PASI50 response stratification (Figure 4) included PASI50 responders demonstrating statistically significant decreases in VEGF-D (−0.09, 95%CI −0.99; 0.82, [*p* = 0.021]), VEGF-A (−0.365 95%CI −2.747; 2.017 [*p* = 0.042]), eotaxin (−0.15 95%CI −1.58 1.27 [0.042]), TARC (−0.92, 95%CI −3.47; 1.62, [*p* = 0.012]), SAA (−1.18, 95%CI −4.40; 2.04, [*p* = 0.004]), MIP-1β (−0.39, 95%CI −2.37; 1.60, [*p* < 0.001]), IL-6 (−0.96, 95%CI −4.18; 2.26, [*p* = 0.012]), IL-27 (−0.187 95%CI −2.13; 1.75 [*p* = 0.016]), IL-22 (−0.62 95%CI −3.52; 2.28 [*p* = 0.042), and CRP (−1.06 95%CI −4.72; 2.61 [*p* = 0.027]) and a statistically significant increase in IL-10 (0.355 95%CI −2.238; 2.80 [*p* = 0.010]) and IL-1α (0.274 95%CI −1.64; 2.91 [*p* = 0.032]) (Appendix A).

#### 2.3.2. Change in Biomarkers in IL-17Ai Responders and Non-Responders (DAPSA50 and PASI50)

Differences in biomarker changes over the 4-month trial period were also evident when comparing IL-17i DAPSA50 responders and non-responders (Figure 4). There were statistically significant decreases in IL-1α (mc −0.13, 95%CI −2.59; 2.32 [*p* = 0.031]) and IL-27 (mc −0.69 95%CI −4.33; 2.94 [*p* = 0.031]), while an increase in IL-17A (2.49, 95%CI −1.86; 6.84, [*p* = 0.031]) was found in DAPSA50 responders. Non-responders revealed statistically significant increases in MIP-3α (0.32, 95%CI −1.02; 1.66, [*p* = 0.016]), IL-8 (0.22, 95%CI −1.76; 2.19, [*p* = 0.007]), IL-17C (0.39, 95%CI −1.09; 1.87, [*p* = 0.010]), IL-7 (0.24, 95%CI −1.71; 2.20, [*p* = 0.003]), IL-17A (2.48, 95%CI −1.46; 6.41, [*p* = 0.009]), VEGF-D (mc 0.66 95%CI −1.20; 1.34 [*p* = 0.027]), and VEGF-C (mc 0.19 (−1.28; 1.66) [*p* = 0.039]) (Appendix A).

PASI50 IL-17Ai responders (Figure 5) demonstrated statistically significant decreases in IL-1α (mc −0.096 (−2.56; 2.37) [*p* = 0.031]), MIP-1β (mc −0.29 95%CI −2.58; 2.01 [*p* = 0.031]), MCP-1 (−0.25 95%CI −2.37; 1.82 [*p* = 0.047]), and eotaxin (mc −0.24 95%CI −1.75; 1.26 [*p* = 0.016]) VEGF-A (mc −0.62 95%CI −2.50; 1.26 [*p* = 0.047]),and increases in IL-17A (3.45 95%CI 0.10; 6.81 [*p* = 0.016]) and PlGF (*p* = 0.047) (Appendix A). There were no statistically significant changes from baseline to follow-up in PASI50 non-responders, probably due to insufficient power (n = 4). For additional biomarkers, it applied to IL-17Ai DAPSA50 responders that most biomarker levels decreased; however, this was not the case for TNFα, IL-5, IL-22, and IFNγ levels which increased, whereas DAPSA50 non-responders increased in biomarker levels. Thus, these results were not statistically significant (Appendix A).

## 3. Discussion

This study evaluated the effect of different medical treatments, TNFi and IL-17Ai, on inflammation-associated biomarkers in a group of PsA patients stratified by the DAPSA50 and PASI50 response to treatment. Biomarker levels in MTX initiators were quantified at baseline to establish biomarker levels in a PsA patient group comprising mainly bio-naïve patients (90%), i.e., medical treatments have had limited effect on the biomarker milieu. Statistically significant differences were only seen in 6 out of 54 biomarkers at baseline when comparing TNFi, IL-17A, and MTX initiators.

The evaluation of biomarkers at baseline revealed differences between patient groups in PsA-associated cytokines, TNFα, and IL-17A. TNFα levels were similar in TNFi and MTX initiators, which might be associated with the majority of patients being bio-naïve. In comparison, TNFα levels displaying a statistically significant increase were demonstrated in IL-17Ai initiators. Higher levels of IL-17A were demonstrated in IL-17Ai initiators compared with both TNFi and MTX initiators. Increased levels of treatment target cytokines TNFα and IL-17A have been associated with previous TNFi treatment, which in other studies has been shown to induce increases in TNFα and IL-17 levels [16,17]. Additional baseline differences between groups include inflammatory cytokines, IL-31 and IL-15, and angiogenesis-associated biomarker Tie-2, which have all been associated with PsA immunopathogenesis [18,19,20]. Nevertheless, it is important to acknowledge that none of the biomarkers measured have been compared with healthy controls, i.e., whether the biomarkers are significantly associated with PsA patients.

Differences in patient characteristics when comparing groups were found in the number of previous bDMARDs, VAS fatigue, and VAS pain, which were all increased in IL-17Ai initiators. This is possibly related to Danish treatment guidelines, including TNFi being the first bDMARD of choice in the PsA treatment. IL-17Ai was typically initiated after TNFi failure. The results implied that included IL-17Ai initiators suffer from higher disease severity or a degree of chronicity, explaining the statistically significant higher VAS pain and VAS fatigue [21], which was supported by the trend indicating longer disease duration for IL-17Ai. None of the clinical and patient-reported outcomes reached a strong correlation, with individual biomarkers reporting the complexity and difficulties of translational examinations finding the link between clinical presentation and PsA immunopathogenesis.

Changes in biomarker levels from baseline to four-month follow-up were evaluated to compare differences between treatment responders and non-responders. It was decided to examine the change in arthritis by DAPSA stratifying by the DAPSA50 response and change in cutaneous psoriasis by PASI stratifying the PASI50 response. Stratification by two clinical symptoms were performed to recognize the sometimes differential effect of treatments in joint arthritis and cutaneous psoriasis [22,23,24].

Previous studies have demonstrated increased cytokine levels, TNFα, IL-17, and IL-10, together with decreases in IL-6 and IL-8 in response to TNFi [16,25], as demonstrated in this study. However, studies exploring changes in response to IL-17Ai remain limited. DAPSA50 responders to TNFi demonstrated a statistically significant increase in IL-10, suggesting the contribution and upregulation of anti-inflammatory properties in responders to TNFi. The increase in IL-10 of responders to treatment is in line with previous data demonstrating an upregulation of IL-10 mRNA expression in response to TNFi [26]. DAPSA50 responders further showed a statistically significant decrease in the pro-inflammatory cytokines IL-6 and CRP and increasing IL-12p70, the last of which is somewhat surprising to previous studies, revealing decreases in pro-inflammatory cytokines that were not the direct target of specific cytokine inhibitors [16]. Decreasing IL-6 levels and the strong trend of decreasing IL-8, most pronounced in PASI50 responders, might imply that TNFi works through the inhibition of the inflammatory amplification process recently proposed [27], which was further supported by the overall decrease in inflammation implied by decreasing CRP. However, increasing IL-17A, IL-1, IL-17B, and IL-17C seems conflicting. Additional statistically significant decreases were seen in acute-phase reactant serum amyloid A [28] and (pro-)inflammatory cytokines IL-27 and IL-22 in responders to TNFi when stratifying by PASI50, suggesting a downregulation of cytokines involved in the recruitment and differentiation of T cells into inflammatory subtypes, such as T helper cell type 1 and 17 (Th1 and Th17) [29,30,31]. Moreover, IL-27 and IL-22 have been considered as important contributors to PsA immunopathogenesis [14]. Increasing TNFα in response to TNFα inhibitor, as demonstrated by previous studies [16,25], was also implied in the current study. However, the increase in TNFα did not reach statistical significance.

PsA patients initiating IL-17Ai revealed statistically significant increases in cytokines IL-8, IL-17C, IL-7, and IL-17A for DAPSA50 non-responders after four months of treatment. Trends of decreasing pro-inflammatory cytokines IL-6, IL-7, and IL-8 were implied in IL-17i DAPSA50 responders, suggesting that IL-17Ai inhibits the amplification process [27]. Interestingly, the same pattern with a statistically significant increase in the target cytokine IL-17A was demonstrated for DAPSA50 and PASI50 responders as well, indicating similar mechanisms with increasing levels in the target cytokine. The increase in target cytokines may be explained by a higher half-life of the cytokines promoted by binding to the TNFi and IL-17Ai, respectively, shown in human subjects treated with TNFi [26]; thus, the bound target cytokine is suggestively inactive [25]. This is supported by studies demonstrating the differential effect on post-treatment TNFα levels depending on generic drugs with different cytokine-binding properties [25]. Further, negative feedback mechanisms have been suggested by the upregulation of TNFα- and IFNγ-positive T cells in response to TNFi [17], which have been contradicted in IL-17Ai initiators by studies revealing the downregulation of mRNA encoding IL-12, IL-17A, IL-21, IL-22, etc. [32]. In this PsA patient cohort, the markedly increased IL-17A after IL-17Ai are also believed to exist as inactive cytokines due to the significant concomitant decrease in pro-inflammatory cytokines IL-8 and IL-6 otherwise induced by IL-17 [27]. However, additional studies are needed to confirm this theory.

The limitations of the study included PsA patients with overall low PASI scores, indicating the need for caution in concluding baseline associations between immune characteristics and cutaneous psoriasis. However, changes over time indicated larger changes, especially in patients experiencing total clearing or exacerbation of cutaneous psoriasis, which is considered interesting in relation to immune response mechanisms. Nonetheless, it is important to consider the possible bias introduced by factors associated with differences in cytokine levels such as BMI, time of day of plasma sample retrieval, etc. Furthermore, stratifying patients further resulted in smaller populations that might reduce the power of the statistical analysis. PsA patients were grouped based on the type of treatment planned for initiation due to the different target cytokines, which suggests different immune response mechanisms. However, 89% of PsA initiating IL-17Ai had been treated with other bDMARDs prior to baseline, which most likely have had an effect on the cytokine milieu. In line with Danish treatment guidelines during the inclusion period, the majority of PsA patients who have received previous bDMARDs received a TNFi. Additionally, several PsA patients were receiving concomitant csDMARDs at a stable dose, which might have an immune modulatory effect as well. Nevertheless, the results are still considered of interest as these patients had high disease activity and the design represents the real-life clinical complexity of PsA patients, and it is the change in cytokine levels in association with response to treatment that remains interesting. Furthermore, it is believed that no stratification will bias the understanding of important immune mechanisms explaining differences between PsA patients and their response to treatment. However, PsA patients were stratified as responders and non-responders based on DAPSA50, which corresponds to a minor improvement in disease activity [33]. PsA patients reaching DAPSA50 will still present an ongoing inflammation possibly reflecting the measured cytokines. It is believed that stratifying by DAPSA70 or DAPSA80 may result in a clearer understanding of treatment response mechanisms in PsA.

## 4. Materials and Methods

### 4.1. Study Participants

Samples to evaluate biomarkers were obtained from the Parker Institute’s clinical observational PsA patient cohort (PIPA), enrolling patients initiating new treatment at Departments of Rheumatology in Region Zealand and the Capital Region of Denmark in accordance with Danish treatment guideline for PsA. Inclusion and exclusion criteria have been published in the PIPA cohort article [34]. An additional inclusion criterion for the current study was added to ensure different treatments including plasma samples from 68 PsA patients initiating TNFi (n = 29), IL-17i (n = 19), or methotrexate (MTX) (n = 20), which were obtained with corresponding clinical and patient-reported data from a baseline visit adjacent to treatment start and after four months of treatment.

### 4.2. Study Visits

The baseline and four-month follow-up visits included clinical examination, blood sampling, and questionnaires collecting relevant patient-reported measures. The clinical examination consisted of a 66/68 swollen and tender joint count, the evaluation of psoriatic skin lesions quantified by the Psoriasis Area Severity Index (PASI), Spondyloarthritis Research Consortium of Canada (SPARCC) enthesitis score, and the examination of dactylitis reported with the number of digits involved. The patient-reported outcome retrieved included the Visual Analog Scale (VAS) patient global, pain, and fatigue. Additionally, the composite measures of Disease Activity Score (DAS28CRP) and Disease Activity in Psoriatic Arthritis (DAPSA) were retrieved to evaluate the response to treatment. Peripheral blood was collected in EDTA vacutainer tubes (Greiner bio-one, Kremsmünster, Germany) and centrifuged to collect plasma. EDTA plasma was transferred and stored at −80 °C until analysis. All procedures were conducted on the same day.

### 4.3. Quantifying Biomarker Levels

The MSD V-plex 54-plex kit (Mesoscale Discovery (MSD), Meso Scale Diagnostic, LCC, Rockville, MD, USA) was used. Patient samples were thawed, prepared in technical doublets, and diluted in-plate. Plate preparation and analysis were conducted in line with the manufacturer’s instructions. Analysis was conducted on the MSD QuickPlex SQ 120 instrument with the software Discovery Workbench version 4.0.12. Biomarker a bsolute values were reported as absolute concentrations in picograms per milliliter (pg/mL).

### 4.4. Statistical Analysis

Before the statistical analysis, biomarker measurements exceeding the allowed intra-assay coefficient of variation (CV) of 30 between doublet samples were excluded. Patient demographics and biomarker levels at baseline were presented as medians with interquartile ranges (IQRs) and numbers with percentages for continuous and categorical variables, respectively. Biomarker levels were presented in pg/mL. The Kruskal–Wallis test was applied at baseline to evaluate differences between treatment groups, i.e., TNFi initiators, IL-17Ai initiators, or MTX initiators. A post hoc Dunn’s test with Bonferroni correction was applied to evaluate statistically significant differences between the individual groups. A Spearman’s rank correlation coefficient (ρ) was included to examine the correlation between (1) individual biomarkers and (2) individual biomarkers and PsA outcomes. A spearman’s value of ρ ≤ 0.29 was considered poor, between 0.3 and 0.59 was considered fair, from 0.6 to 0.8 was considered moderately strong, and >0.8 was considered very strong [15]. To compare responders and non-responders to treatment, PsA patients were stratified based on the DAPSA50 and PASI50 response (yes/no), i.e., whether an improvement of at least 50% in DAPSA and PASI score was achieved from baseline to follow-up. The primary analysis was performed on DAPSA50 responders as not all included PsA patients presented with active psoriasis at baseline. QQ plots applied to evaluate for normal distribution. Biomarker levels were log-transformed, and the change over time was calculated as mean changes with standard deviations to be visualized in bar charts. Biomarkers with >10% missing were excluded from Spearman’s ρ and the mean change bar chart. A Wilcoxon signed-rank test was used to examine differences between paired measures: clinical, patient-reported, and biomarker levels. It applied to all analyses, and *p* < 0.05 was considered statistically significant. The statistical analysis was carried out using the statistical software *R* with additional relevant packages.

## 5. Conclusions

This study aimed to assess the effect of TNFi and IL-17Ai on various biomarkers, revealing interesting results regarding immune treatment response mechanisms. The results demonstrated the importance of studying immune response mechanisms in different well-defined subgroups of PsA to improve the understanding of PsA clinical heterogeneity and different responses to treatments.

## Figures and Tables

**Figure 1 ijms-25-03002-f001:**
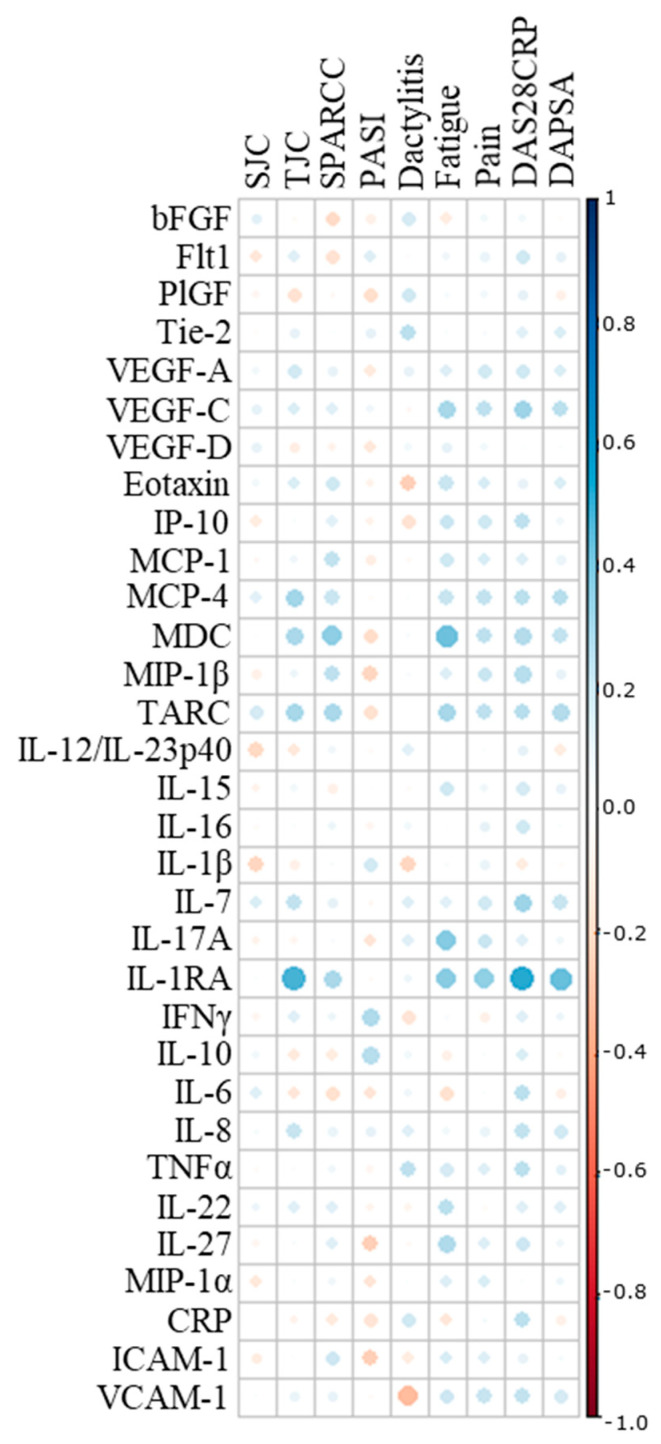
Correlation analysis. Spearman’s correlation coefficient was implemented to assess associations between individual biomarkers and clinical and patient-reported outcomes. SJC, swollen joint count; TJC, tender joint count; SPARCC, Spondyloarthritis Research Consortium of Canada; PASI, Psoriasis Area Severity Index; DAS28CRP, Disease Activity Score calculated with 28 joints and c-reactive protein; DAPSA, disease activity in psoriatic arthritis; bFGF, basic fibroblast growth factor; Flt-1, Fms-related receptor tyrosine kinase-1; PlGF, placental growth factor; Tie-2, endothelial receptor tyrosine kinase; VEGF, vascular endothelial growth factor; IP-10, IFN-induced protein-10; MCP, monocyte chemoattractant protein; MDC, macrophage-derived chemokine; MIP, macrophage inflammatory protein; TARC, thymus and activation-regulated chemokine; IL, interleukin; IL-1RA, interleukin-1 receptor antagonist; IFN, interferon; TNF, tumor necrosis factor; CRP, c-reactive protein; ICAM, intercellular adhesion molecule; VCAM, vascular cell adhesion molecule.

**Figure 2 ijms-25-03002-f002:**
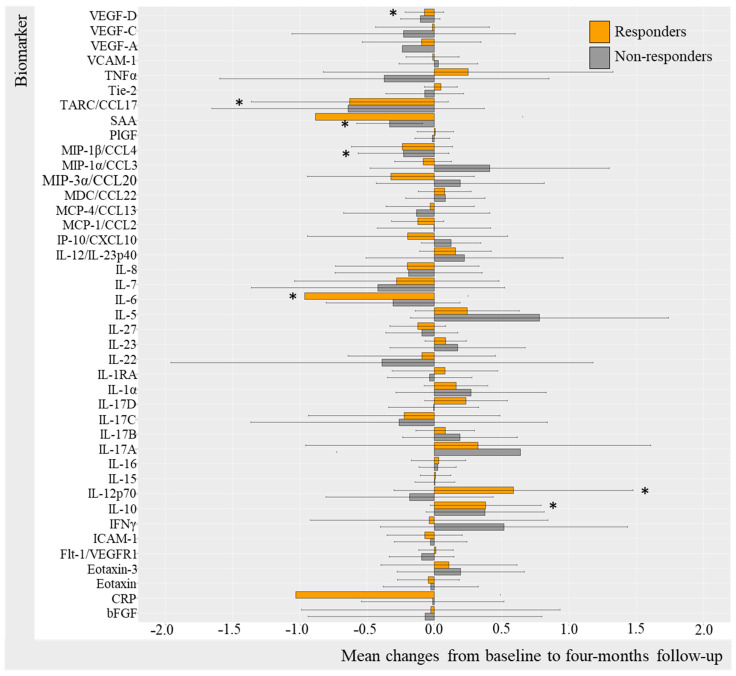
Mean changes in biomarker levels in TNFi initiators stratified by DAPSA50. Log-transformed mean changes from baseline to four-month follow-up stratified by DAPSA50 corresponding to a 50% improvement in DAPSA, included as a measure of arthritic joint disease (Appendix A). * (star) represents changes that are statistically significant. TNFi, tumor necrosis factor alpha inhibitor; DAPSA, disease activity in psoriatic arthritis; VEGF, vascular endothelial growth factor; VCAM, vascular cell adhesion molecule; TNF, tumor necrosis factor; Tie-2, endothelial receptor tyrosine kinase; TARC, thymus and activation regulated chemokine; CCL, CC chemokine ligand; SAA, serum amyloid A; PlGF, placental growth factor; MIP, macrophage inflammatory protein; MDC, macrophage-derived chemokine; MCP, monocyte chemoattractant protein; IP-10, IFN-induced protein-10; CXCL, CX chemokine ligand; IL, interleukin; IL-1RA, interleukin-1 receptor antagonist; IFN, interferon; ICAM, intercellular adhesion molecule; Flt-1, Fms-related receptor tyrosine kinase-1; CRP, c-reactive protein; bFGF, basic fibroblast growth factor.

**Figure 3 ijms-25-03002-f003:**
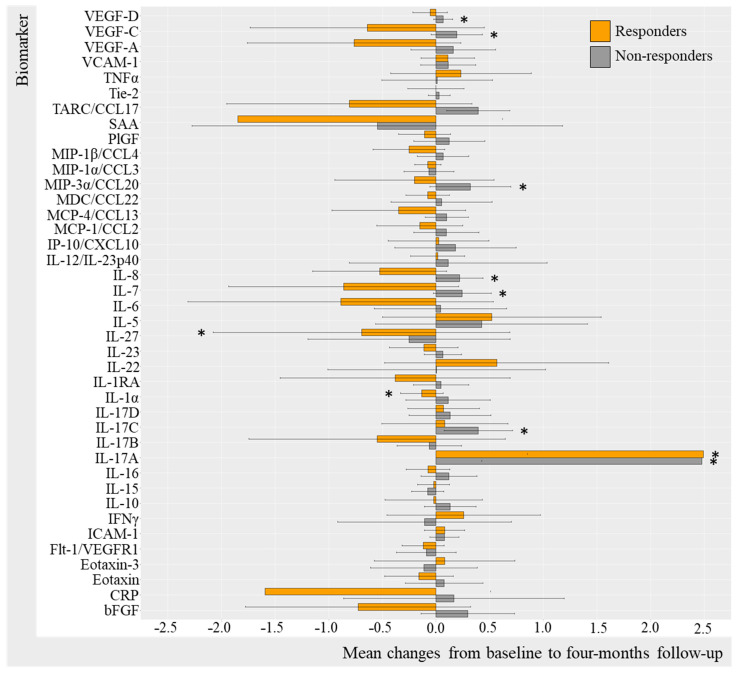
Mean changes in biomarker level in IL-17Ai initiators stratified by DAPSA50. Log-transformed mean changes from baseline to four-month follow-up stratified by DAPSA50 corresponding to a 50% improvement in DAPSA, included as a measure of arthritic joint disease (Appendix A). * (star) represents changes that are statistically significant. IL-17Ai, interleukin-17A inhibitor; DAPSA, disease activity in psoriatic arthritis; VEGF, vascular endothelial growth factor; VCAM, vascular cell adhesion molecule; TNF, tumor necrosis factor; Tie-2, endothelial receptor tyrosine kinase; TARC, thymus and activation regulated chemokine; CCL, CC chemokine ligand; SAA, serum amyloid A; PlGF, placental growth factor; MIP, macrophage inflammatory protein; MDC, macrophage-derived chemokine; MCP, monocyte chemoattractant protein; IP-10, IFN-induced protein-10; CXCL, CX chemokine ligand; IL, interleukin; IL-1RA, interleukin-1 receptor antagonist; IFN, interferon; ICAM, intercellular adhesion molecule; Flt-1, Fms-related receptor tyrosine kinase-1; CRP, c-reactive protein; bFGF, basic fibroblast growth factor.

**Figure 4 ijms-25-03002-f004:**
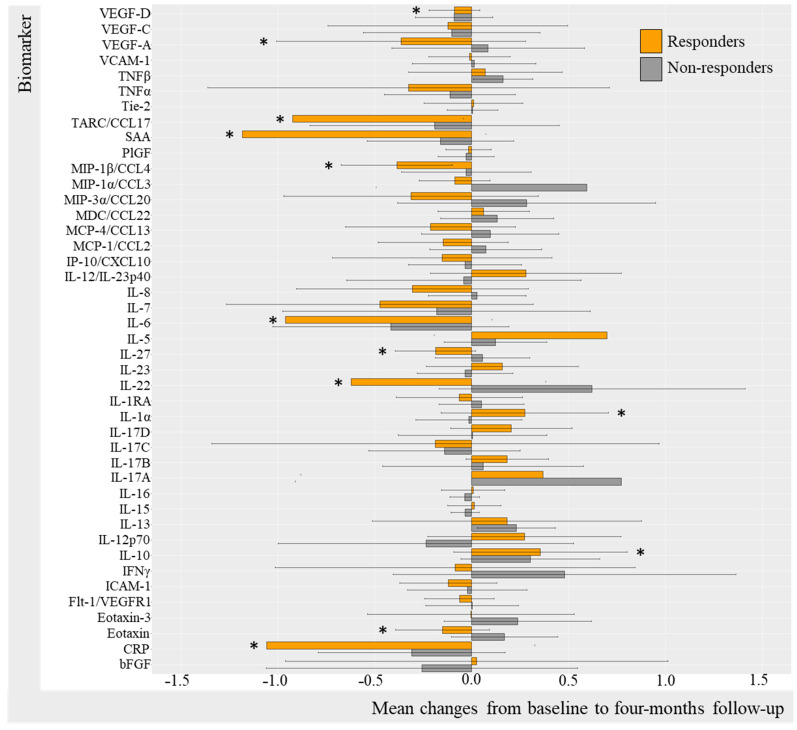
Mean changes in biomarker levels in TNFi initiators stratified by PASI50. Log-transformed mean changes from baseline to four-month follow-up stratified by PASI50 corresponding to 50% improvement in PASI included as a measure of cutaneous psoriasis (Appendix A). * (star) represents changes that are statistically significant. TNFi, Tumor Necrosis Factor alpha inhibitor; PASI, Psoriasis Area Severity Index; VEGF, Vascular Endothelial Growth Factor; VCAM, vascular cell adhesion molecule; TNF, Tumor Necrosis Factor; Tie-2, endothelial receptor tyrosine kinase; TARC, Thymus and activation regulated chemokine; CCL, CC chemokine ligand; SAA, serum amyloid A; PlGF, Placental Growth Factor; MIP, macrophage inflammatory protein; MDC, macrophage-derived chemokine; MCP, monocyte chemoattractant protein; IP-10, IFN-induced protein-10; CXCL, CX chemokine ligand; IL, interleukin; IL-1RA, interleukin-1 receptor antagonist; IFN, interferon; ICAM, intercellular adhesion molecule; Flt-1, Fms-related receptor tyrosine kinase-1; CRP, c-reactive protein; bFGF, basic fibroblast growth factor.

**Figure 5 ijms-25-03002-f005:**
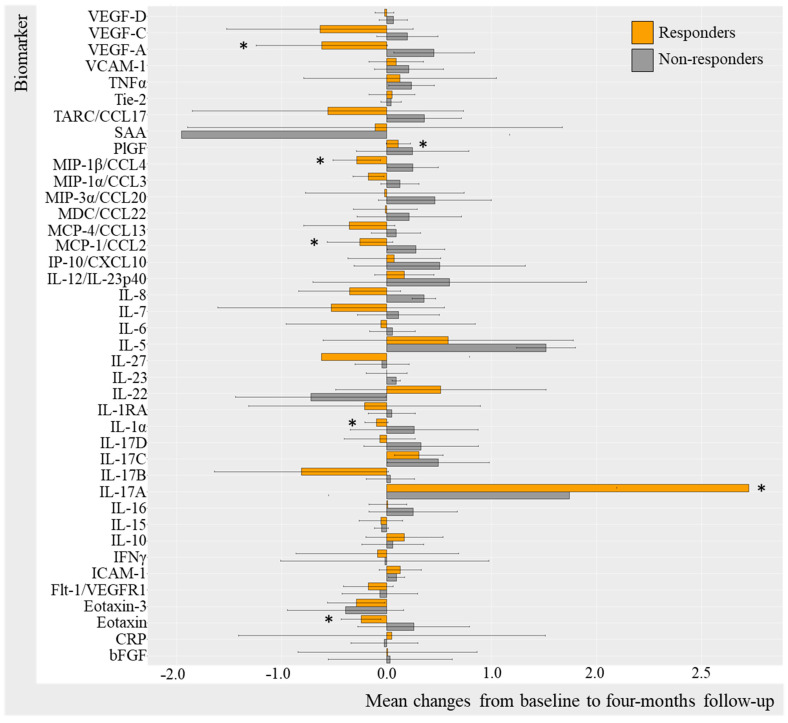
Mean changes in biomarker level in IL-17Ai initiators stratified by PASI50. Log-transformed mean changes from baseline to four-month follow-up stratified by PASI50 corresponding to 50% improvement in PASI included as a measure of cutaneous psoriasis (Appendix A). * (star) represents changes that are statistically significant. IL-17i, interleukin-17 inhibitor; PASI, Psoriasis Area Severity Index; VEGF, Vascular Endothelial Growth Factor; VCAM, vascular cell adhesion molecule; TNF, Tumor Necrosis Factor; Tie-2, endothelial receptor tyrosine kinase; TARC, Thymus and activation regulated chemokine; CCL, CC chemokine ligand; SAA, serum amyloid A; PlGF, Placental Growth Factor; MIP, macrophage inflammatory protein; MDC, macrophage-derived chemokine; MCP, monocyte chemoattractant protein; IP-10, IFN-induced protein-10; CXCL, CX chemokine ligand; IL, interleukin; IL-1RA, interleukin-1 receptor antagonist; IFN, interferon; ICAM, intercellular adhesion molecule; Flt-1, Fms-related receptor tyrosine kinase-1; CRP, c-reactive protein; bFGF, basic fibroblast growth factor.

**Table 1 ijms-25-03002-t001:** Baseline characteristics.

	All (n = 68)	TNFi Initiators (n = 29)	IL17Ai Initiators (n = 19)	MTX Initiators (n = 20)	*p* Value
Female, n (%)	38 (55.88)	16 (55.17)	12 (63.16)	10 (20.00)	0.707
Age, years	51.50 (45.27–60.88)	49.60 (56.20–59.30)	49.80 (45.05–61.10)	56.80 (51.35–62.38)	0.173
Disease duration, years	4.00 (1.58–10.00)	4.00 (1.75–8.42)	6.00 (1.63–14.5)	4.73 (1.44–10.75)	0.794
bDMARD monotherapy, n (%)	27 (56.25) ^§^	17 (58.62)	10 (52.63)	-	0.911
Methotrexate, n (%)	15 (31.25) ^§^	10 (34.48) ^§^	5 (26.32) ^§^	-	0.319
Leflunomide, n (%)	4 (8.33) ^§^	1 (3.45) ^§^	3 (15.79) ^§^	-	
Salazopyrin, n (%)	2 (4.17) ^§^	1 (3.34) ^§^	1 (5.26) ^§^	-	
Previous bDMARD, n (%)					
- 0	39 (57.35)	19 (65.52)	2 (10.53)	18 (0.90)	**<0.001**
- 1	14 (20.59)	7 (24.14)	5 (26.32)	2 (0.10)	
- 2	8 (11.77)	2 (5.90)	6 (31.58)	0 (0.00)	
- ≥3	7 (10.29)	1 (3.45)	6 (31.58)	0 (0.00)	
DAPSA	29.20 (21.90–38.85) ^†^	28.80 (22.00–39.80)	33.25 (25.80–46.65) ^†^	27.55 (20.50–36.83)	0.277
DAS28CRP	4.12 (3.70–4.90) ^†^	4.12 (3.58–4.85)	4.36 (3.77–5.30) ^†^	4.06 (3.61–4.57)	0.274
PASI (0–72)	1.20 (0.00–3.08)	1.50 (0.30–3.40)	0.40 (0.00–2.00)	0.95 (0.00–2.93)	0.453
VAS fatigue (0–100 mm)	67.00 (50.00–82.00) ^†^	60.00 (48.00–79.00)	85.00 (75.00–91.50)	62.00 (46.50–75.50)	**0.007**
VAS pain (0–100 mm)	59.00 (31.00–78.00) ^†^	57.00 (27.00–77.00)	75.00 (50.75–83.75)	47.00 (31.50–64.25)	0.176

Patient and treatment characteristics were presented as medians with a corresponding interquartile range (IQR) for continuous variables and a number with a corresponding percentage for categorical variables. ^§^ Percentage of patients receiving bDMARD (n = 48). ^†^ Data are missing for one patient. bDMARD, biological disease-modifying anti-rheumatic drug; DAPSA, disease activity for psoriatic arthritis; DAS28CRP, Disease Activity Score calculated by a 28 tender/swollen joint assessment and c-reactive protein; PASI, Psoriasis Area Severity Index; US, ultrasonography; VAS, Visual Analog Scale.

**Table 2 ijms-25-03002-t002:** Change in clinical outcomes of DAPSA50 and PASI50 responders and non-responders to TNFi and IL-17Ai.

	Baseline	Follow-Up	Mean Change	*p*-Value
**DAPSA50 Responders**				
**TNFi (n = 11)**				
DAPSA	35.80 ± 23.76	7.05 ± 9.48	−28.75 (−81.81; 24.32)	**<0.001**
PASI	2.16 ± 2.72	0.56 ± 1.22	−1.60 (−7.79; 4.59)	**0.024**
**IL17Ai (n = 7)**				
DAPSA	33.93 ± 14.01	12.28 ± 7.18	−21.65 (−55.95; 12.65)	**0.031**
PASI	1.51 ± 1.88	0.21 ± 0.45	−1.30 (−5.45; 2.85)	0.106
**PASI50 Responders**				
**TNFi (n = 12)**				
DAPSA	36.58 ± 23.70	15.53 ± 17.80	−21.04 (−82.21; 40.13)	**<0.001**
PASI	4.33 ± 5.55	0.25 ± 0.60	−4.08 (−15.61; 7.45)	**0.003**
**IL17Ai (n = 7)**				
DAPSA	38.40 ± 11.06	26.30 ± 17.06	−12.1 (−56.41; 32.21)	0.156
PASI	5.14 ± 8.37	0.17 ± 0.34	−4.97 (−22.95; 13.00)	**0.016**
**DAPSA50 Non-responders**				
**TNFi (n = 9)**				
DAPSA	34.73 ± 13.08	27.58 ± 15.59	−7.16 (−49.90; 35.59)	**0.039**
PASI	4.56 ± 6.22	5.09 ± 8.14	0.53 (−20.99; 22.05)	1.000
**IL17Ai (n = 12)**				
DAPSA	38.53 ± 17.87	39.68 ± 22.31	1.15 (−57.84; 60.14)	0.850
PASI	3.70 ± 7.32	1.95 ± 4.60	−1.75 (19.59; 16.09)	0.834
**PASI50 Non-responders**				
**TNFi (n = 6)**				
DAPSA	33.03 ± 10.28	20.65 ± 15.33	−12.38 (−52.61; 27.84)	0.094
PASI	2.13 ± 2.44	8.17 ± 8.68	6.03 (−13.61; 25.68)	0.094
**IL17Ai (n = 4)**				
DAPSA	34.20 ± 14.26	34.15 ± 21.98	−0.05 (−60.47; 60.37)	1.000
PASI	4.75 ± 5.65	5.85 ± 6.86	1.10 (−19.38; 21.58)	0.414

Mean change in clinical outcome from baseline to four-month follow-up including PsA patients with follow-up data and complete data to retrieve the composite measures. Response to treatment was defined as a 50% improvement in DAPSA/PASI from baseline to follow-up. DAPSA, disease activity in psoriatic arthritis; PASI, psoriasis area severity index; TNFi, tumor necrosis factor alpha inhibitor; IL17i, interleukin-17 inhibitor.

## Data Availability

The dataset supporting the conclusions of this article cannot be shared publicly due to the privacy of the individuals who participated in the study. Data may be shared as part of research collaborations between participating institutions in line with GDPR and if approved by the Parker Institute and Danish authorities.

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
