# Peer review of "Changes in Inflammatory Cytokines in Responders and Non-Responders to TNFα Inhibitor and IL-17A Inhibitor: A Study Examining Psoriatic Arthritis Patients"

_ijms, 2024, doi:10.3390/ijms25053002_

Round 1
Reviewer 1 Report
Comments and Suggestions for Authors
I completed the revision of the manuscript entitled „Changes in inflammatory cytokines in responders and non-responders to Tumor Necrosis Factor α inhibitor and IL-17A inhibitor; a study examining Psoriatic Arthritis patients”. The structure of the manuscript (Introduction, Methodology, Results, Discussion) follows a logical sequence. Introduction contains enough background informations. The study is well designed and conducted, but patients should be better described, especially for those qualified for biologic treatment (what drugs they were treated with before, including biologics).
Major revision:
1. In Table 1, the last column gives a p value without specifying which study groups it applies to. It should be specified exactly between which groups (TNFi initiators, IL-17Ai initiators or MTX initiators) statistical significance was shown.
2. In Table 1, bDMARDs are given only in monotherapy, add what drugs were used in the others (n=12 TNFi initiators; n=9 17Ai initiators). MTX? or other biologic drugs?
3. In view of the above, it is puzzling why these people qualified for the study, and how can the effect of treatment with biologic inhibitors be known when other drugs were also treated?
4. Statistical significance (p value) should be plotted on Figures 1 and 2.
Minor revision:
1. Line 194 - should be added (Table S3).
2. Item 198 - should be replaced (Figure S1 to Figure 3) - error.
3. In addition, Figures S1 and S2 should be in the text and not in the Supplementary Materials.

Author Response
1. In Table 1, the last column gives a p value without specifying which study groups it applies to. It should be specified exactly between which groups (TNFi initiators, IL-17Ai initiators or MTX initiators) statistical significance was shown.
Author reply: Thank you for the comment. We have added a clarification (page 3, line 91-94) to the table text to support the method section.
2. In Table 1, bDMARDs are given only in monotherapy, add what drugs were used in the others (n=12 TNFi initiators; n=9 17Ai initiators). MTX? or other biologic drugs?
Author reply: We appreciate the suggestion and have added the number of patients in treatment with concomitant csDMARD, including the corresponding percentage (page 3, table 1)
3. In view of the above, it is puzzling why these people qualified for the study, and how can the effect of treatment with biologic inhibitors be known when other drugs were also treated?
Author reply: We thank reviewer for the question and acknowledge that concomitant csDMARD might have an effect on the immune response mechanisms in the patients of the cohort. However, patients were in stable csDMARD dose at study baseline at which bDMARD was initiated due to the high disease activity, meaning that we can assume that improved symptoms after initiation of the bDMARD (while being on stable csDMARD dose) can be attributed to the bDMARD and change in cytokines caused by the bDMARD intervention. Nevertheless, acknowledge the question we have added to the discussion section as a limitation (page 12, line 348 – page 13, line 358).
4. Statistical significance (p value) should be plotted on Figures 1 and 2.
Author reply: We thank reviewer for the suggestion and have indicated the statistically significant changes in all figure with a star.
Minor revision:
1. Line 194 - should be added (Table S3).
Author action point: A reference to supplementary tables have been added to all figures 2-5, line 177, 186, 196 and 211
2. Item 198 - should be replaced (Figure S1 to Figure 3) - error.
Author action point: We acknowledge the suggestion to change the figure. However, as the study originated from a Dept. of rheumatology, it was decided to prioritize the inclusion of figures visualizing change associated with change in arthritic joint symptoms, i.e., DAPSA, in the main manuscript. Additional action point see below the next comment.
3. In addition, Figures S1 and S2 should be in the text and not in the Supplementary Materials.
Author action point: We thank reviewer for the suggestion. Indeed, we find it relevant to include all figures in the main text. If the increasing number of tables can be granted in accordance with journal guidelines. All figures have been moved to the main text as figure 4 and 5 (page 8 and 9).

Reviewer 2 Report
Comments and Suggestions for Authors
The manuscript entitled “Changes in inflammatory cytokines in responders and non-responders to Tumor Necrosis Factor α inhibitor and IL-17A inhibitor; a study examining Psoriatic Arthritis patients” by Skougaard M et al is an original research that investigates the effects of TNF and IL-17 inhibitors on different biomarkers in psoriatic arthritis patients. The manuscript provided interesting data and it is well written. However, there are some points to clarify:
Major Comments
1. In reference to study participants, it is not clear how each treatment is assigned to patients. Is it random? Please explain.
2. What are the exclusion and inclusion criteria? Given that different infectious and metabolic diseases (such as overweight or obesity) produce changes in circulating cytokines levels, it is very important to clarify and/or analyse if differences observed by authors are produced only by PsA.
3. In Table 1 please add units. It is not clear between which groups there are significant differences.
4. Format of Table 2 is confused.
5. Please mention which normality test was used to analyse data distribution.
6. Is data presented between line 186 -193 represented in any figure or table? If so, please indicate.
7. It would be very important for authors to mention which the previous bDMARD patients received before TNF or IL17 inhibitor treatment.
8. Line 198. Authors referred IL-17i DAPSA50 data to figure S1, but this figure shows: “Mean changes in biomarker levels in TNFi initiators stratified by PASI50”. Please verify.

Author Response
Reviewer 2:
The manuscript entitled “Changes in inflammatory cytokines in responders and non-responders to Tumor Necrosis Factor α inhibitor and IL-17A inhibitor; a study examining Psoriatic Arthritis patients” by Skougaard M et al is an original research that investigates the effects of TNF and IL-17 inhibitors on different biomarkers in psoriatic arthritis patients. The manuscript provided interesting data and it is well written. However, there are some points to clarify:
Major Comments
1. In reference to study participants, it is not clear how each treatment is assigned to patients. Is it random? Please explain.
Author response: Patients were included from Departments of Rheumatology (standard care), why treatment was assigned in accordance with Danish treatment guidelines for PsA. Clarification has been added page 13, line 369-373.
2. What are the exclusion and inclusion criteria? Given that different infectious and metabolic diseases (such as overweight or obesity) produce changes in circulating cytokines levels, it is very important to clarify and/or analyse if differences observed by authors are produced only by PsA.
Author response: A reference to in- and exclusion criteria have been added on page 13, line 371-374, excluding infections. We acknowledge that several factors produce changes in cytokine levels, and may further add simple factors such as time during the day that the samples were obtained, female hormonal cycles etc. Unfortunately, we cannot provide the detailed information necessary to conduct the analysis of high quality, which have been added to the limitation section (page 12, line 345-347)
3. In Table 1 please add units. It is not clear between which groups there are significant differences.
Author response: We thank for the comment and have added relevant units, page 3, table 1
4. Format of Table 2 is confused.
Author response: We acknowledge the comment and have done our best to update the table to make it less confusing (page 5)
5. Please mention which normality test was used to analyse data distribution.
Author response: We thank reviewer for the recommendation. A clarification has been added to the method section, page 14, line 417
6. Is data presented between line 186 -193 represented in any figure or table? If so, please indicate.
Author response: Data presented between line 186-193 (original submission) are figure text. No changes have been made. If reviewer meant to point out data other lines, it is most welcome.
7. It would be very important for authors to mention which the previous bDMARD patients received before TNF or IL17 inhibitor treatment.
Author response: Unfortunately, complete data is not available to include this information. However, during the time of inclusion, patients have mainly been treated with TNFi previous to initiation TNFi or IL-17Ai studied as this would be considered the Danish treatment guidelines during the inclusion period. This have been added to the discussion section, page 12, line 345-347
8. Line 198. Authors referred IL-17i DAPSA50 data to figure S1, but this figure shows: “Mean changes in biomarker levels in TNFi initiators stratified by PASI50”. Please verify.
Author response: Thank you for the comment. Additional updated references have been added to section 2.3.1 and 2.3.2

Round 2
Reviewer 1 Report
Comments and Suggestions for Authors
There is one more point to improve.
A statistically significant result of the Kruskal-Wallis test only tells us that at least one of the groups is different from another group. Therefore, we then perform post-hoc tests (usually the Dunn or Dunn test with Bonferroni correction) to check exactly which groups differ from each other. What post-hoc test did the authors use and between which groups was there statistical significance (all? or perhaps between TNFi initiators and IL-17Ai initiators? or TNFi initiators and MTX initiators? or IL-17Ai initiators and MTX initiators? - should be completed in the table 1. Therefore, the explanations added by the authors (page 3, lines 91-94) are unnecessary, because they still do not characterize which three groups are statistically significant.
Author Response
Reviewer 1:
A statistically significant result of the Kruskal-Wallis test only tells us that at least one of the groups is different from another group. Therefore, we then perform post-hoc tests (usually the Dunn or Dunn test with Bonferroni correction) to check exactly which groups differ from each other. What post-hoc test did the authors use and between which groups was there statistical significance (all? or perhaps between TNFi initiators and IL-17Ai initiators? or TNFi initiators and MTX initiators? or IL-17Ai initiators and MTX initiators? - should be completed in the table 1. Therefore, the explanations added by the authors (page 3, lines 91-94) are unnecessary, because they still do not characterize which three groups are statistically significant.
Author response:
We thank reviewer for the suggestion. No post hoc analysis was applied to the original paper, but we acknowledge the need. Statistical analyses have been updated, including a Dunn's test with Bonferroni correction both for table 1 and supplementary table S1. The method section has been updated page 15, line 413-415. Additionally, the result section corresponding to table 1 can be found on page 2, line 74-79, and supplementary table S1, page 2, 81-88.
